# OpenReview forum: "AutoBaxBuilder: Bootstrapping Code Security Benchmarking"
_ICML.cc/2026/Conference — ICML 2026 regular_

### Official Review · Reviewer_JUnc · 2026-03-11

**Soundness:** 2
**Presentation:** 3
**Significance:** 3
**Originality:** 2
**Overall Recommendation:** 4
**Confidence:** 3

**Summary:**

This paper proposes AutoBaxBuilder, an LLM-based pipeline for automatically generating security-oriented code-generation benchmark tasks from scratch, including scenario specifications, functional tests, and end-to-end exploits. The method is expected to work as a way to extend BAXBENCH-style evaluation while reducing manual effort and mitigating benchmark aging/contamination. The paper validates the generated tests/exploits by comparing against expert-written BAXBENCH artifacts on 28 existing scenarios, then uses the pipeline to build AutoBaxBuilder, a 40-scenario benchmark with easy/medium/hard splits. The reported results suggest that the generated benchmark is challenging for current models and that the construction cost is low.

**Compliance With Llm Reviewing Policy:**

Affirmed.

**Final Justification:**

the rebuttal has addressed most of my concerns.

**Key Questions For Authors:**

1. While the authors claim the frameowork as model- and framework-agnostic in principle, what's the implication of its reliance on Python-FastAPI reference solutions?

**Limitations:**

yes

**Strengths And Weaknesses:**

1. The paper addresses a relevant and timely problem. Benchmark construction for secure code generation is expensive, and the motivation to automate scenario, test, and exploit generation is compelling.
2. The empirical validation against BAXBENCH looks good. The authors report that functional-test outcomes match BAXBENCH on 80.9% of solutions, and they manually audit 71 generated security tests with only one unsound exploit.
3. The benchmark construction effort seems practically useful. The paper claims roughly 12x reduction in human effort and low per-scenario API cost, which makes the contribution potentially valuable as an evaluation infrastructure paper.
---
1. The reported evidence sometimes cuts both ways. For example, the paper emphasizes that AUTOBAXBUILDER flags many more solutions as insecure than BAXBENCH, which may indeed indicate better exploit coverage. But it could also reflect over-aggressive or benchmark-specific exploit generation. So, if I understand correctly, it mainly depends on the chosen CWEs?

2. The novelty is good at the systems/evaluation level, but the paper is lighter on deeper scientific insight. The work is primarily a pipeline plus validation study (to a degree, it somewhat looks like a technical report). That can still be publishable, but then I would expect especially strong evidence of robustness, generality, and external validity. Right now, I found the paper useful and well executed, but not yet fully convincing at that level.

---

> ### Author Rebuttal · Authors · 2026-03-31
>
> We thank the reviewer for their feedback, highlighting the practical relevance and significance of our findings. Below, we address the remaining concerns.
> > Q1: What implication does the choice of reference solution framework have on the constructed benchmark?
>
> Python-FastAPI was chosen as a pragmatic default: current models are proficient at it, and it provides useful logging for the pipeline's feedback signal.
>
> To directly ablate the sensitivity to this choice, we regenerate the test cases and exploits using JavaScript-Fastify and Go-Gin as reference solution frameworks. The same set of LLM-generated solutions was then evaluated against all three benchmark suites.
>
> Model rankings and CWE coverage remain stable, and numbers do not strongly deviate per chosen backend, confirming that the benchmark's discriminative power is not an artifact of the reference framework. We report the full sec_pass@1 (pass@1) below.
>
>
> |Framework|Sonnet 4.5|Gemini 3 Pro|Sonnet 3.7|Gemini 2.5 Pro|Grok 4|GPT-4o|Codestral|Qwen2.5 72B|Llama3.3 70B|Qwen2.5 7B|
> |-|-|-|-|-|-|-|-|-|-|-|
> |Python-FastAPI|44% (81%)|43% (74%)|34% (68%)|31% (55%)|30% (61%)|16% (41%)|10% (31%)|5% (16%)|5% (18%)|2% (5%)|
> |JavaScriptFastify|44% (87%)|42% (82%)|34% (71%)|33% (61%)|30% (64%)|15% (40%)|10% (32%)|5% (16%)|6% (20%)|2% (5%)|
> |Go-Gin|48% (82%)|52% (82%)|36% (71%)|38% (58%)|33% (59%)|20% (37%)|14% (27%)|8% (15%)|9% (17%)|3% (5%)|
>
>
> We also compare CWE coverage across reference frameworks. For each CWE, we report the fraction of correct solutions flagged as vulnerable.
>
> |CWE|Python-FastAPI|JavaScript-Fastify|Go-Gin|
> |-|-|-|-|
> |20|48%|48%|34%|
> |22|6%|3%|3%|
> |78|0%|5%|0%|
> |79|14%|13%|12%|
> |89|1%|0%|0%|
> |94|0%|4%|0%|
> |284|2%|3%|1%|
> |434|4%|7%|11%|
> |522|4%|0%|3%|
> |703|3%|0%|6%|
> |863|4%|3%|3%|
>
> The dominant CWEs, CWE-20 (Improper Input Validation) and CWE-79 (XSS), are consistently flagged across all three benchmarks, highlighting the generality of AutoBaxBuilder. CWE-78 (OS Injection) and CWE-94 (Code Injection) are caught in Fastify because Node.js supports `child_process.execSync()` for the `compiler` scenario, and Go-Gin covers CWE-434 (Dangerous Upload) better in `pdf_to_text` due to its more permissive multipart handling.
>
> Overall, we observe reference language artifacts to have marginal influence. This is partially due to our pipeline being explicitly designed to mitigate framework-specific assumptions. For example, functional tests use status code categories over specific return values or singular status codes (`P_{Func Reqs}`, App D.2). RefineTests (Algorithm 4 and App D.2) specifically regularizes over-specific assertions. For instance, in the SVGBadge case study (App C.1), an exact-match test was generalized to a property-based check after implementations differed in (underspecified) attribute ordering.
>
> > Q2: Why does AutoBaxBuilder have both more and less coverage than BaxBench?
>
> We refer the reviewer to our answer to Q1 of reviewer *iWNz*.
>
> > Q3: Can you highlight some additional, more generalizable or rigorous insights for future research?
>
> We list three key insights for future research below.
>
> - *Benchmarks can be dynamically created*: AutoBaxBuilder’s key contribution is enabling dynamic benchmark creation. By continuously generating fresh tasks with tunable difficulty, it directly addresses contamination and saturation problems of prior benchmarks. Beyond static evaluation, due to the low cost and scalability, it is viable to integrate into RL training, wherein the pipeline can provide an uncontaminated reward signal for improving a model’s default security.
> - *Self-bias can be circumvented*: A key distinction from settings where self-preference bias is typically present (e.g., LLM-as-a-judge [1]) is that the orchestration LLM in our pipeline is grounded in black-box execution feedback. The LLM does not infer outputs. Additionally, we take preventive measures to limit information flow across pipeline steps by design (e.g., withholding solution code during parts of test refinement, Section 3) to discourage test-solution co-adaptation. Our ablation (Figure 8, App B.2) confirms empirically that models do not score differently on benchmarks they helped construct.
> - *Models are capable of writing sound exploits*: Our approach yields 96.8% soundness in our human evaluation (Table 4, App B.5) and only 1 unsound exploit out of 71 on AutoBaxBuilder-constructed BaxBench. Our ablations confirm similar results. This insight can enable the design of security-first agents. Additionally, the hardened solutions can be used as a training signal for security-first code generation.
>
> **References**
>
> [1] https://arxiv.org/abs/2306.05685

---

> > ### Author Rebuttal · Reviewer_JUnc · 2026-04-03
> >
> > Overall, the rebuttal is pretty good.Howevr, my concern in Q1 is improved but not fully settled. The use of human disagreement explains why human evaluation is noisy, but it also somewhat reduces confidence in human audit as a gold standard.

---

> > > ### Author Response · Authors · 2026-04-03
> > >
> > > We thank the reviewer for acknowledging the strength of our rebuttal and for adjusting their score accordingly. We will include the limitation of our human rater experiment explicitly in the next revision.

---

### Official Review · Reviewer_wEj9 · 2026-03-12

**Soundness:** 2
**Presentation:** 3
**Significance:** 3
**Originality:** 3
**Overall Recommendation:** 5
**Confidence:** 4

**Summary:**

This paper proposes an automated strategy for generating security benchmarks called AutoBaxBuilder in a 3-step process. In particular, they use an orchestrator LLM which proposes a task and some sample implementations in YAML format. Then, functional tests are generated which triggers refinement of the initial solutions. Finally, they use an exploit generation step to create insecure and secure version with supplied exploits. Using this approach, they can drastically scale up the creation of security benchmarks. Their results show that new benchmarks generated through AutoBaxBuilder have similar results to the original BaxBench dataset.

**Compliance With Llm Reviewing Policy:**

Affirmed.

**Final Justification:**

I liked this paper a lot. Security benchmarking is an important problem, and the authors seemed to develop good infrastructure for generating new benchmarks. While there might be some minor issues such as coverage, I think it is much better than any other solution we currently have. Practically speaking, If an LLM performed well on a lot of AutoBaxBuilder generated datasets I would trust it much more than if it didn't perform well, and if it didn't perform well I would be very skeptical using it to generate a web application. Seems to me like the main thing you would want from a benchmark.

The rebuttal changed my evaluation of the paper a lot (I was originally a 3, now am a 5). My main concern was that it felt weird to use LLM's to generate benchmarks for LLM's - it seems to me a bit like grading your own test. The authors performed some ablations/additional experiments that convinced me it's not a problem in practice.

Overall, I liked this paper, and recommend its acceptance.

**Key Questions For Authors:**

My main question from the Weakness section: If the LLM's being tested were given the ability to test exploits and generate exploits against their solution, just like the IdentifyVulnerabilities and RefineExploit steps, wouldn't they be able to get 100% on the benchmarks?

**Limitations:**

Yes

**Strengths And Weaknesses:**

#### Strengths
I think this is a good approach, and a very important problem. Benchmarking security on LLM's is an under-studied task in my opinion, and the results are definitely promising.

#### Weaknesses
~I have one concern with the paper. I think something that feels a little off to me is that the LLM's that we are benchmarking are also the ones that are generating the dataset, which doesn't sit quite right with me. If the LLM's being tested were given the ability to test exploits and generate exploits against their solution, just like the IdentifyVulnerabilities and RefineExploit steps, wouldn't they be able to get 100% on the benchmarks? This is the *only* real limitation for me, but it's a big one because i feel like it almost negates the whole purpose of having a benchmark in the first place. I'd be very interested to hear the author's thoughts on this.
My best guess is that in practice, people aren't going to be using this pipeline when they're just generating code or vibe coding, so the ability to measure the inherent security of LLM-generated code is important.~
EDIT: This has been addressed. I still think it's a bit of an issue, but I think in practice it's much better than any alternative I can think of. Honestly, nowadays if we tried to generate a human-written benchmark, they would probably still use an LLM.

---

> ### Author Rebuttal · Authors · 2026-03-31
>
> We thank the reviewer for recognizing the practical importance of this problem and the value in our approach. We address their core concern in three sub-questions below.
>
> > Q1: Is it a problem to use LLMs to construct evaluations for their own outputs?
>
> We do not believe this is inherently problematic. A key distinction from settings where self-preference bias is typically present (e.g., LLM-as-a-judge [1]) is that the orchestration LLM in our pipeline is grounded in black-box execution feedback. The LLM does not infer outputs. Additionally, we take preventive measures to limit information flow across pipeline steps by design (e.g., withholding solution code during test refinement, Section 3) to discourage test-solution co-adaptation.
>
> A question that remains is whether models exhibit systematic bias on benchmarks they helped construct, which we address next.
>
> > Q2: Is there a bias of LLMs testing against their own solutions?
>
> We had this concern ourselves and for this reason included an ablation on the choice of LLMs in Appendix B.2. In it, we use a disjoint set of LLMs for constructing a *variant of AutoBaxBench*. We call the variant of AutoBaxBench we generated “Ablation”. We are then interested in whether a model exhibits a bias in performance on the benchmark it helped construct. To inspect this, we plot the performance of all evaluated LLMs on both the original AutoBaxBench constructed in Section 4.2 and the Ablation variant against each other in Figure 8. The high rank correlation ($\rho > 0.9$) between the scores on AutoBaxBench and its Ablation variant indicates that no, in fact, there is no strong bias of models performing better on benchmarks in which they were used for construction.
>
>
> > Q3: Could LLMs score 100% on AutoBaxBench with the right tooling?
>
> We thank the reviewer for this interesting idea. (1) We investigated the effect of harnesses and the ability to test their own generations and found surprisingly little improvement. (2) Meanwhile, we argue that even if a specialized scaffold would achieve a very high score on our benchmark, it would not contradict our findings about the security of LLMs in typical usage.
>
> (1) We investigated the effect of using agentic harnesses in a realistic environment. Concretely, we tasked GPT-5 using the Codex harness and Claude Sonnet 4.5 in the Claude Code harness to generate solutions for BaxBench. In the initial prompt, we directed the agent to write their own tests to verify functionality and security of its solution [2], and equipped the harness with standard tool use for multi-turn execution.  For a fair comparison, we use the one-shot baseline with a security prompt [3], comparable to the instructions given to the agentic scaffold. We evaluate on both the AutoBaxBuilder-generated BaxBench suite and the original BaxBench suite and report sec_pass@1 (pass@1), excluding CWE-400 for consistency with the paper.
>
> | AutoBaxBuilder | One-shot | Harness |
> | --- | --- | --- |
> | Sonnet 4.5 | 54% (78%) | 49% (77%) |
> | GPT-5 | 48% (62%) | 57% (75%) |
>
> | BaxBench | One-shot | Harness |
> | --- | --- | --- |
> | Sonnet 4.5 | 70% (80%) | 54% (81%) |
> | GPT-5 | 52% (57%) | 59% (74%) |
>
> Notably, we observe that agentic harnesses do not come close to 100%. While Claude Sonnet 4.5 does not exhibit an improvement at all, there is a clear improvement of GPT-5. This pattern is consistent across AutoBaxBuilder and human-written BaxBench, and aligns with the findings of [4], in which they found only marginal, if any, improvements in functionality and security through agentic scaffolds.
>
> (2) It is conceivable that an LLM in a *specialized* scaffold could achieve consistently higher scores on AutoBaxBench than an LLM without additional scaffold as evaluated in our work. However, we argue that this does not contradict the goal of our benchmark; as the reviewer pointed out themselves, AutoBaxBench sets out to measure the security of LLMs in typical usage, and such scaffolds are unlikely to be employed in typical usage.
>
> **References**
>
> [1] https://arxiv.org/abs/2306.05685
> [2] This prompt for the agent is added to the standard prompt: "Write your own tests to verify the correctness and security of your implementation."
> [3] This generic security prompt is: "Make sure to follow best practices for security and avoid common vulnerabilities."
> [4] https://arxiv.org/abs/2502.11844

---

> > ### Author Rebuttal · Reviewer_wEj9 · 2026-03-31
> >
> > I see. It's quite surprising to me that Claude performance doesn't get better with the harness. But either way, I'm raising my score to recommend acceptance. I think it's a good paper.
> >
> > I think the concern about LLM's generating benchmarks to test LLM's is something that should have a fairly lengthy discussion in the appendix or something, because it's definitely a concern to me still. However, I don't really have a better solution so I think it's fine. IMHO having a benchmark with potential flaws is WAY better than having no benchmark at all.
> >
> > One other thing: How do you envision this being used in practice? Is the workflow something like, I run `./autobaxbuilder.sh`, that generates a dataset basically the same as BaxBench (obviously different exploits/stuff but roughly the same tasks), and then I test my LLM on that benchmark? Just want to be super clear.

---

> > > ### Author Response · Authors · 2026-04-01
> > >
> > > We thank the reviewer for their valuable feedback and updated score. We will add a dedicated discussion about LLM’s generating benchmarks to test LLMs in the next revision, as suggested.
> > >
> > > On the practical workflow: that is exactly right. We provide a convenience script that wraps AutoBaxBuilder’s Python CLI and generates multiple scenarios, with associated tests and exploits, in parallel. One can tune the difficulty, reference framework, and choice of orchestration and solution LLMs, or keep our pragmatic defaults.
> > >
> > > The output is a fresh BaxBench-compatible benchmark instance, consisting of scenarios with tests and exploits, which you can evaluate any LLM on. Additionally, logs, verdicts, intermediate solutions and intermediate tests/exploits are stored, and can conceivably be used for e.g., security-aware training.

---

### Official Review · Reviewer_iWNz · 2026-03-13

**Soundness:** 3
**Presentation:** 3
**Significance:** 3
**Originality:** 2
**Overall Recommendation:** 4
**Confidence:** 4

**Summary:**

This paper proposes AUTOBAXBUILDER, an LLM-based agentic pipeline that automatically generates code security benchmark tasks from scratch. Each task consists of a
  web backend scenario (OpenAPI specification), functional tests, and end-to-end security exploits — following the design of BAXBENCH, a recent human-crafted
  benchmark. The pipeline operates in three steps: (1) scenario generation, (2) functional test construction with iterative refinement against reference solutions,
  and (3) exploit generation with contrastive solution pairs to ensure exploits are sound.

**Compliance With Llm Reviewing Policy:**

Affirmed.

**Final Justification:**

This is overall a good paper that provides a way to construct baxbench-like data. I think my only concern is that we maybe able to provide much better benchmark that is comparable to AI's capability in discovering deeply hidden, long-standing vulnerabilities. But this is out of the scope of this paper so I still give weak accept as my recommendation.

**Key Questions For Authors:**

- Q1. How does AUTOBAXBUILDER compare to SEC-bench [1]? SEC-bench also automates security benchmark construction at lower cost ($0.87 vs. $3.90 per instance). What
  are the qualitative differences in the generated benchmarks, and can you provide a direct comparison on overlapping CWE categories?

 - Q2. The expert evaluation rates exploit coverage at 71% (Table 4). What is the main bottleneck — does the orchestration LLM fail to identify the vulnerability,
  fail to generate a working exploit, or fail to construct contrastive solutions?

 - Q3. How sensitive are the generated tests and exploits to the choice of reference solution framework? If the pipeline were run with Go/Gin or Node/Express
  reference solutions instead of Python-FastAPI, would the resulting benchmarks produce different model rankings?

-  Q4. For the BAXBENCH validation, can you report results separately for models with training cutoffs before vs. after BAXBENCH's publication (February 2025)?

  [1] Li et al. "SEC-bench: Automated Benchmarking Framework for Security Engineering Tasks." NeurIPS 2025. arXiv:2506.11791.

**Limitations:**

Mostly yes. The paper discusses scope (REST APIs only), difficulty scaling, and challenging vulnerability types (CWE-400, plaintext credentials). However, it does not discuss:
- Potential data leakage from BAXBENCH being in training data of evaluation models,
- The Python-FastAPI reference solution bias and its effect on cross-framework fairness

**Strengths And Weaknesses:**

##  S1. Well-motivated problem.
Human-crafted security benchmarks like BAXBENCH are expensive and don't scale. As LLMs improve and  benchmarks contaminate training data, the ability to continuously generate fresh, high-quality security tasks is genuinely valuable.

##  S2. Useful benchmark artifact (Significance).
AUTOBAXBENCH with three difficulty tiers provides a challenging, extensible evaluation resource. The Hard subset  (sec_pass@1 = 25% for the best model) leaves substantial room for future improvement.

##  W1. Incomplete related work.
Several closely related works on automated security benchmarking are not discussed:
  - SEC-bench [1] is a fully automated security benchmark construction pipeline at $0.87 per instance.  This paper is very similar so I am concerned about novelty.
  - DualGauge [2] automates joint security-functionality benchmarking with 154 tasks paired with both functional and security tests.
  - CVE-Bench [3] provides a sandbox framework for LLM agents to exploit vulnerable web applications, closely related to the web backend focus here.

##  W2. Reference solutions only in Python-FastAPI.
The pipeline constructs all reference solutions in Python-FastAPI (Section 4.1), but evaluates LLMs across 14
  frameworks and 6 languages. Tests and exploits designed against Python-FastAPI behavior may encode framework-specific assumptions (e.g., default error handling, MIME parsing, parameter coercion) that penalize correct implementations in other frameworks.

##  W3. Exploit coverage is low. Expert evaluation (Table 4) rates exploit coverage at only 71.07%, meaning ~30% of exploits may not generalize to new implementations.
   Combined with 164 false negatives where BAXBENCH detected vulnerabilities that AUTOBAXBUILDER missed (Section B.4), this suggests the pipeline's exploit
  generation is substantially less comprehensive than human experts, particularly for CWE-522 (credential inspection) and nuanced attack vectors.

##  W4. Potential data leakage in BAXBENCH validation. The validation (Section 4.2) compares AUTOBAXBUILDER-generated tests against human-written BAXBENCH tests.
  BAXBENCH was publicly available since February 2025 (GitHub + arXiv), while some models have later training cutoffs — notably Claude 4 Sonnet (solution LLM, cutoff
   March 2025) and Claude 4.5 Sonnet (evaluated, cutoff July 2025). The ρ = 0.93 correlation could be partially inflated by memorization. The paper discusses
  contamination from benchmark construction (Section 4.4) but not from BAXBENCH itself being in training data.

  [1] Li et al. "SEC-bench: Automated Benchmarking Framework for Security Engineering Tasks." NeurIPS 2025. arXiv:2506.11791.

  [2] "DualGauge: Automated Joint Security-Functionality Benchmarking for Secure Code Generation." arXiv:2511.20709, November 2025.

  [3] "CVE-Bench: A Benchmark for AI Agents' Ability to Exploit Real-World Web Application Vulnerabilities." ICML 2025 (spotlight). arXiv:2503.17332.

  [4] Ullah et al. "CVE-Genie: From CVE Entries to Verifiable Exploits." arXiv:2509.01835, September 2025.

---

> ### Author Rebuttal · Authors · 2026-03-31
>
> We thank the reviewer for their feedback, highlighting the relevance and significance of our work. Below, we address and clarify the reviewer's concerns in detail.
> > Q1: Why does AutoBaxBuilder have more or less coverage than BaxBench?
>
> We note that the benchmark is designed to provide a security upper bound, flagging vulnerabilities that are certain to exist. Despite the 71% coverage from human evaluation (Table 4), AutoBaxBuilder produces a more difficult benchmark than the human-written baseline (Figure 4). The number itself should be interpreted with care: inter-rater agreement is only 23%, indicating that human experts themselves disagree on when a given exploit's payloads are sufficiently thorough.
>
> We explain differences in coverage through identifiable gaps.
>
> *More coverage*: AutoBaxBuilder’s pipeline maximizes the attack surface, considering vulnerabilities in each of the solutions. The RefineExploit loop rotates through multiple reference solutions, naturally broadening exploit diversity. This yields more thorough exploits than BaxBench in 39% of scenarios (e.g., SVG and HTML payloads for XSS in ImageTransfer vs. only HTML in BaxBench, App B.4) and exploits entirely new CWEs that BaxBench doesn’t test for in 21% of scenarios (e.g., OS injection in FileSearch, App B.4).
>
> *Less coverage*: We more closely analyze the 164 false negatives in BaxBench. We find that only 9% are never considered by the orchestration LLM. The majority (74%) is discarded during refinement. The pipeline discards exploits that fail to correctly differentiate secure from insecure solutions after at most 5 iterations, with an average of 1.4 exploits discarded per scenario (App B.1). The remaining 18% passed filtering but were not triggered by the specific vulnerable code in the tested solution.
>
> The concentration on CWE-522 (Figure 11) is explained by BaxBench's catch-all rule for probing a fixed list of insecure passwords, which is by default enabled for all scenarios. Our pipeline does not generate such generic tests across scenarios.
> > Q2: What implication does the choice of reference solution framework have on the constructed benchmark?
>
> We refer the reviewer to our answer to Q1 of reviewer *JUnc*.
> > Q3: Is there any possible contamination from the BaxBench data and what does the 0.93 correlation Figure 8 show?
>
> We are confident that contamination through BaxBench is not an issue for the following reasons.
>
> First, golden solutions to BaxBench scenarios were never published [1], eliminating leakage during evaluation or reference solution construction. Second, all 40 scenarios of AutoBaxBench were generated entirely from scratch, ruling out contamination in those.
>
> The remaining question is whether tests and exploits generated for BaxBench scenarios (Section 4.2) were contaminated by the published originals. Note that the test cases and exploits are generated by the orchestration LLM, GPT-5. The model's knowledge cutoff is 30th September 2024 [2], predating BaxBench’s release (February 2025 [1]), and has no internet access during pipeline execution. This further rules out contamination for the test and exploit generation.
>
> Note that Figure 8 (App B.2), where we present  $\rho=0.93$, does not compare model generated tests to BaxBench tests. We compare model performance on AutoBaxBench vs. an ablated variant, constructed using a disjoint set of LLMs. The high rank correlation confirms model rankings to be consistent across independently constructed benchmarks. Notably, there is no systematic bias of models performing better on benchmarks in which they were used for construction.
>
> > Q4: Please discuss the following related work
>
> SEC-bench [3] reconstructs benchmarks from existing known CVEs, allowing them to achieve a significantly lower per-instance cost. AutoBaxBuilder constructs complete tasks from scratch. Moreover, SEC-bench focuses solely on memory vulnerabilities: the set of CWEs they consider is orthogonal from ours.
>
> DualGauge [4] requires human experts to steer test/exploit refinement and evaluates isolated functions rather than end-to-end web scenarios. CVE-Bench [5] evaluates LLMs’ offensive capabilities against real-world vulnerable applications, complementary to our focus on security of LLM-generated code. Like SEC-bench, it is anchored in real-world CVEs, reintroducing the contamination risk AutoBaxBuilder is designed to avoid.
>
> We will extend our existing discussion of SEC-Bench in the submitted paper and add the other prior works in the next revision.
>
> **References**
>
> [1] https://github.com/logic-star-ai/baxbench/releases
> [2] https://developers.openai.com/api/docs/models/gpt-5
> [3] https://arxiv.org/abs/2506.11791
> [4] https://arxiv.org/abs/2511.20709
> [5] https://arxiv.org/abs/2503.17332

---

> > ### Author Rebuttal · Reviewer_iWNz · 2026-04-03
> >
> > Thanks for the rebuttal. All of my concerns have been resolved. I will raise my score.
> > However, I am still quite interested in one thing: how to generate data of sufficiently high quality. Recent work has shown that AI is already capable of discovering deeply hidden, long-standing vulnerabilities in widely used projects — for instance, Anthropic's recent discovery of vulnerabilities in Linux kernel that had gone undetected for years. This suggests that AI should, in principle, also be capable of generating vulnerabilities of comparable depth and subtlety, which could be used to amplify vulnerability data for training and human learning. However, as of now, I have not seen any AI-generated dataset or benchmark that reaches this level of quality.

---

> > > ### Author Response · Authors · 2026-04-04
> > >
> > > We thank the reviewer for their careful assessment and are glad to have been able to resolve their concerns.
> > >
> > > This is an interesting observation and one we share enthusiasm for. The gap between AI’s ability to find deep vulnerabilities in existing code and its ability to generate scenarios that require equally subtle security reasoning is a notable asymmetry. Benchmarks rooted in real-world CVEs (e.g., SEC-Bench [1], CVE-bench [2]) already capture difficult - but known and disclosed - vulnerabilities through patches of existing codebases. However, generating *novel* code scenarios from scratch that would naturally elicit vulnerabilities of comparable depth and subtlety is a challenge.
> > >
> > > Our hope is that AutoBaxBuilder can inspire future work to re-use its concepts such as contrastive refinement to push towards agentic pipelines generating scenarios on repository scale.
> > >
> > > We thank the reviewer for the interesting point raised, and we will add a discussion of this direction to the outlook section in the revised manuscript.​​​​​​​​​​​​
> > >
> > > [1] https://arxiv.org/abs/2506.11791
> > > [2] https://arxiv.org/abs/2503.17332

---

### Decision · Program_Chairs · 2026-04-30

**Decision:**

Accept (regular)

**Comment:**

This submission addresses an important practical problem and presents a benchmark construction pipeline that reviewers found useful overall. The main positives are the ability to scale benchmark generation and the reasonably strong validation against BAXBENCH. At the same time, reviewers also noted limits in exploit coverage and in how much confidence can be placed in the human audit. The rebuttal resolved several of the main concerns, including questions about contamination, self-bias, and framework dependence